# Green Nanocomposites from Rosin-Limonene Copolymer and Algerian Clay

**DOI:** 10.3390/polym12091971

**Published:** 2020-08-30

**Authors:** Hodhaifa Derdar, Geoffrey Robert Mitchell, Vidhura Subash Mahendra, Mohamed Benachour, Sara Haoue, Zakaria Cherifi, Khaldoun Bachari, Amine Harrane, Rachid Meghabar

**Affiliations:** 1Centre de Recherche Scientifique et Technique en Analyses Physico-Chimiques (CRAPC), BP 10 384, Siège ex-Pasna Zone Industrielle, Bou-Ismail CP 42004, Tipaza, Algeria; hodhaifa-27@outlook.fr (H.D.); zakaria.cherifi.17@gmail.com (Z.C.); bachari2000@yahoo.fr (K.B.); 2Laboratoire de Chimie des Polymères (LCP), Département de Chimie, FSEA, Oran1University Ahmed Benbella BP N 1524 El M’Naouar, Oran 31000, Algeria; med-073@live.fr (M.B.); rachidmeghabar@yahoo.fr (R.M.); amineharrane@yahoo.fr (A.H.); ritedj1991@hotmail.fr (S.H.); 3Centre for Rapid and Sustainable Product Development, Institute Polytechnic of Leiria, 2430-082 Marinha Grande, Portugal; vidhumahendra@googlemail.com; 4School of Chemistry Food Science and Pharmacy, University of Reading, Reading RG6 6AD, UK; 5Department of Chemistry, FSEI University of Abdelhamid Ibn Badis—Mostaganem, Mostaganem 27000, Algeria

**Keywords:** green nanocomposites, rosin, organoclay, Maghnite-CTA^+^

## Abstract

Green nanocomposites from rosin-limonene (Ros-Lim) copolymers based on Algerian organophilic-clay named Maghnite-CTA^+^ (Mag-CTA^+^) were prepared by in-situ polymerization using different amounts (1, 5 and 10% by weight) of Mag-CTA^+^ and azobisisobutyronitrile as a catalyst. The Mag-CTA^+^ is an organophilic montmorillonite silicate clay prepared through a direct exchange process; the clay was modified by ultrasonic-assisted method using cetyltrimethylammonuim bromide in which it used as green nano-filler.The preparation method of nanocomposites was studied in order to determine and improve structural, morphological, mechanical and thermal properties ofsin.The structure and morphology of the obtained nanocomposites(Ros-Lim/Mag-CTA^+^) were determined using Fourier transform infrared spectroscopy, X-ray diffraction, scanning electronic microscopy and transmission electronic microscopy. The analyses confirmed the chemical modification of clay layers and the intercalation of rosin-limonene copolymer within the organophilic-clay sheets. An exfoliated structure was obtained for the lower amount of clay (1% wt of Mag-CTA^+^), while intercalated structures were detected for high amounts of clay (5 and 10% wt of Mag-CTA^+^). The thermal properties of the nanocomposites were studied by thermogravimetric analysis (TGA) and show a significant improvement inthe thermal stability of the obtained nanocomposites compared to the purerosin-limonene copolymer (a degradation temperature up to 280 °C).

## 1. Introduction

In the most recent decades, interest has increased in a new class of materials, which consist of one or more distinct components that together produce nanomaterials called nanocomposites. The use of these new materials was initiated by Toyota researchers in the early 1990s. In fact, by dispersing clays in polyamide-6 by in-situ polymerization, they showed a significant improvement in the dimensional stability of the polymer matrix [1]. These results have been perspectives for polymer matrix nanocomposites in many scientific areas [2]. The use of a polymer matrix by adding a well-defined percentage of clay as a nano-reinforcing filler leads to the improvement of the physicochemical properties of the obtained nanocomposites, such as excellent stability, good chemical resistance, high mechanical resistance, high degradation temperature and increased biodegradability of biodegradable polymers [3,4]. In recent years, some nanocomposites based on toxic polymers have been replaced by others based on green materials. Nanocomposites based on polymers and clay can be obtained by different methods such as in-situ polymerization, molten state or solution blending of components [5]. Based on the synthesis method and the type of the polymer matrix, two different types of nanocomposite structures can be obtained: intercalated and exfoliated nanocomposites [6].

Rosin is yellowish-brown solid form of resin obtained from pine and similar types of trees belonging to the conifer family, produced by heating the liquid resin from the tree to vaporize the volatile liquid terpene components [7]. The pine resin collected by tapping the tree approximately contains 70% rosin, 15% turpentine and 15% debris and water, as shown in Scheme 1 [8]. Rosin is composed of rosin acid isomers from the diterpene group. The particular composition depends on the plant species and the country of origin from which the rosin is extracted. Due to the cheap cost of production of rosin, the high availability and its unique properties (hydrophobicity, biocompatibility and chemical reactivity), rosin is used in a wide range of products such as emulsifier [9], water proofing agent and insulating material [10]. However, its potential applications as advanced materials are somewhat limited due to certain properties of rosin, which includes the low softening point (70 °C), the weak mechanical properties (brittleness at room temperature) and the high acidity. Therefore, various attempts for the chemical modification of rosin have been reported [11,12].

Limonene is a monocyclic terpene found in many essential oils extracted from citrus zest [13]. The first polymerization of terpenes was carried out in 1798, when Bishop Watson added a drop of sulphuric acid to produce a sticky resin [14]. The copolymerization of limonene with other monomers such as styrene was also attempted using azobisisobutyronitrile (AIBN) as a radical catalyst and with β-pinene using Maghnite-H^+^ as a cationic catalyst [15]. In 1950, William Roberts studied the cationic polymerization of many terpenes such as limonene, α-pinene and β-pinene using Lewis acids as catalysts, for example, aluminum trichloride (AlCl_3_) and tin tetrachloride (SnCl_4_), by adding less than 1% of the catalyst. William Roberts produced a solid β-pinene polymer with a low molecular weight of about 1500 g/mol [16]. Derdar et al. also studied the polymerization of limonene using a green catalyst called Maghnite [17]. The degree of polymerization of β-pinene, although very low, is higher than that of the two other monomers (limonene, α-pinene) [18,19]. Limonene has been widely used in a wide range of products such as cosmetics, food additives, medicine and even as a green solvent.

The use of organoclay (Mag-CTA^+^) has an impact on the morphology of the prepared nanocomposites, especially in the dispersion of in-situ polymerization methods. For these reasons, several nanocomposites based on polymers and organoclay were prepared [20,21,22,23,24]. By reviewing the literature, we found that no previously published work has described the use of natural clay as a nano-reinforcing filler in the preparation of green-nanocomposites based on rosin-limonene copolymer. In this study, after modification of clay by CTAB: cetyltrimethylammonuim bromide, we developed the synthesis of green nanocomposites Ros-Lim/Mag-CTA^+^ by in-situ polymerization method. In our study, a solution reaction was used in order to have uniform heating and also to avoid transfer reactions at a temperature between 85 and 90 °C using AIBN as a radical initiator. Our results were confirmed with different characterization methods such as Fourier Transform Infrared Spectroscopy (FTIR), X-ray diffraction analysis (XRD), scanning electronic microscopy (SEM), transmission electronic microscopy (TEM) and thermogravimetric analysis (TGA).

## 2. Materials and Methods

### 2.1. Materials 

In this work, we used gum rosin (obtained via local producers, Costa e Irmãos, Leiria, Portugal). (R)-(+)-limonene (97%), methanol (CH_3_OH, 99.9%), toluene (C_6_H_5_CH_3_, 99.8%), sodium chloride (NaCl), azobisisobutyronitrile (AIBN, 98%) and cetyltrimethylammonuim bromide (CTAB) were purchased by (Sigma Aldrich, Leiria, Portugal) and used as received. Maghnite (Algerian montmorillonite) was supplied in the raw state by ENOF Bental Spa of the National Company of Nonferrous Mining Products, Maghnia Unit (Maghnia, Algeria).

### 2.2. Preparation of the Organoclay (Mag-CTA^+^)

The preparation of organoclay (Mag-CTA^+^) consists of the exchange of Mag-Na^+^ (clay exchanged with sodium) by a cationic surfactant [25]. Maghnite is Algerian montmorillonite sheet silicate clay with very high ratio of silicon and aluminium, and has an interfoliar space of 1.01 nm in the raw state [26]. We prepared the organoclay from Mag-Na^+^ using a concentration of cationic exchange capacity (1 CEC) of CTAB (CEC = 90 meq/g). First, 10 g of Mag-Na^+^ was placed in a 1L Erlenmeyer flask with the chosen concentration (1CEC); the activation of the organoclay was carried out by ultrasound for 1 h [20]. At the end of the exchange process, the suspension was filtered and then washed several times with distilled water. Finally, the solid obtained was dried at 105 °C for 24 h and ground. The structure of organo-phyllic clay was confirmed by FT-IR and XRD analysis, and their morphological properties were studied by SEM and TEM analysis.

### 2.3. Preparation of Green Nanocomposites (Ros-lim/Mag-CTA^+^)

Ros-Lim/Mag-CTA^+^ nanocomposites were prepared using in-situ polymerization method. 0.03 mol (9 g) of rosin and 0.03 mol (5 g) of limonene were dissolved in toluene, and then different amounts of organoclay (1, 5 and 10% by weight) were added to the mixture. The solution was then stirred for 5 min to completely dissolve the rosin and limonene in toluene. After that, the radical initiator azobisisobutyronitrile was added to the mixture and stirred (500 rpm) under reflux at 85–90 °C for 6 h (See Scheme 2). At the end of the reaction, the nanocomposite was recovered by precipitation in cold methanol, filtered and dried under vacuum overnight. The operating conditions are shown in Table 1.

### 2.4. Characterization

The functional groups of the modified clay and the obtained nanocomposites were analyzed by infrared spectroscopy (FT-IR) in the range of 4000–360 cm^−1^ using Bruker ALPHA ALPHA Diamond-ATR (France). The morphology of the modified clay and its nanocomposites were observed by XRD diffraction patterns using a Bruker AXS D8 diffractometer (Cu-K radiation) (Germany); FEG-SEM on a JEOL 7001F (Tokyo, Japan) electron microscopy and transmission electron microscopy were performed using a Hitachi 8100 (Tokyo, Japan). Thermal properties were analyzed by thermogravimetric analysis (TGA) using a PerkinElmer STA 6000 (Waltham, MA, USA) under nitrogen in the temperature range of 30–700 °C with a heating rate of 20 °C/min. 

## 3. Results

### 3.1. Modified Clays (Mag-Na^+^ and Mag-CTA^+^)

The FT-IR spectra of Mag-Na^+^ and Mag-CTA^+^ are shown in Figure 1. We observe an intense peak at 1057 cm^−1^ and two bands at 455 and 515 cm^−1^ relating to the elongation vibrations of the Si–O–Si and Si–O–Al bonds, respectively [27,28]. The band at 1000 cm^−1^ is due to the vibration of Si–O of the Maghnite. Different bands were obtained after the modification of Maghnite by CTA^+^, and two new bands were observed for Mag-CTA^+^, in the 2850 and 2922 cm^−1^ regions corresponding to the C–H stretching vibrations of the methyl group. The results obtained by FT-IR analysis confirm the intercalation of the alkyl ammonium ions of the CTAB between the clay sheets.

The X-ray diffractograms (Figure 2) of Raw-Mag, Mag-Na^+^ and Mag-CTA^+^ show that the calculated basal spacing (d001), applying Bragg equation (2.d.sinθ = n.λ), is 1.01 nm for Raw-Mag and 1.29 nm for Mag-Na^+^. This increase in basal spacing is explained by the substitution of sodium between the sheet of Raw-Mag. The diffractogram of Mag-CTA^+^ shows that the interfoliar distance goes from d = 1.29 nm for the Mag-Na^+^ to d = 1.8 nm for the Mag-CTA^+^. This increase indicates that there is intercalation of the alkyl ammonium ions of the CTAB in the inter-foliar galleries. The addition of the alkyl ammonium ions causes a displacement of the characteristic peak of the interfoliar distance of montmorillonite towards the weak angles (4.90°) for Mag-CTA^+^. Aicha Khenif et al. [25] obtained an interlayer distance of 1.98 nm during 24 h of stirring for the preparation of CTAB/Clay; in our case, an interlayer distance of 1.8 nm was obtained only in 1 h.

### 3.2. Obtained Nanocomposites (Ros-Lim/Mag-CTA^+^)

The FT-IR spectra of the obtained nanocomposites (Ros-Lim/Mag-CTA^+^ 1, 5 and 10%) and pure Ros-Lim copolymer are shown in Figure 3. The infrared spectra show that the copolymer and the nanocomposites have almost the same vibration bands overlapping with the vibration bands of the organo-modified clay (Mag-CTA^+^), and the FT-IR spectra show that the obtained nanocomposites Ros-Lim/Mag-CTA^+^ are in a good agreement with the Ros-Lim structure. The adsorption band at 1690 cm^−1^ corresponds to the bond C=O of rosin acid that was observed in the FT-IR spectra of the obtained nanocomposites. The C-H symmetric and asymmetric stretching of the methyl and methylene group were observed at 2922 and 2935 cm^−1^. Compared with the FT-IR spectrum of the pure copolymer, the spectra of the obtained nanocomposites show the appearance of the intense peak at 1000 cm^−1^ corresponding to the vibration of Si–O of the Mag-CTA^+^. However, the absorption bands observed on the FT-IR spectra of Ros-Lim and Mag-CTA^+^ are gathered on the FT-IR spectra of the obtained nanocomposites. These results show the intercalation of Ros-Lim copolymer in the interlayer montmorillonite gallery.

The XRD patterns of the obtained copolymer and green-nanocomposites are shown in Figure 4. The XRD pattern of Ros-Lim copolymer presents no sharp peak confirming its amorphous structure. This amorphous structure is observed in the nanocomposites with the appearance of additional peak characteristic of Mag-CTA^+^, which confirms its good dispersion in the copolymer matrix. The nanocomposites prepared by (5 and 10%) of Mag-CTA^+^ showed a single peak around 2θ = 2 and 3° corresponding to the interlayer distances d_001_= 4.15 and 3.3 nm, respectively. The interlayer distance of these nanocomposites was increased more than twice compared to the Mag-CTA^+^, which had an interlayer distance of 1.8 nm. This result also confirms that the copolymer was well intercalated between the clay galleries. Except for the case of nanocomposites obtained by 1 % of Mag-CTA^+^, the diffraction peak of Mag-CTA^+^ was not observed, confirming the exfoliation of the clay, whichexplains a good diffusion of the Ros-Lim copolymer in the clay galleries. These results are in agreement with those obtained by HanèneSalmi-Mani et al. [29]. 

Figure 5 shows the SEM images of the organo-modified clay (Mag-CTA^+^), the intercalated nanocomposites (Ros-Lim/Mag-CTA^+^ 5 and 10%) and the exfoliated nanocomposites (Ros-Lim/Mag-CTA^+^ 1%). The comparison of the Mag-CTA^+^ morphology (Figure 5e) with the first nanocomposites Ros-Lim/Mag-CTA^+^ 10 and 5% (Figure 5b,c) and the pure copolymer Ros-Lim (Figure 5a) shows a more organized small particle structure of montmorillonite. In the nanocomposites Ros-Lim/Mag-CTA^+^ 1% (Figure 5d), the observation of nanocomposites at 10 μm reveals that a formation of montmorillonite plate separated, that is, a partial exfoliation, the same sample observed at 2 μm, shows a rougher surface and a covering of the montmorillonite surface by the copolymer.

The transmission electron microscopy was used to determine the dispersion of Mag-CTA^+^ in Ros-Lim copolymer matrix and also to compare the results obtained by the XRD analysis. The TEM images of Mag-CTA^+^ and the obtained nanocomposites Ros-Lim/Mag-CTA^+^ (1, 5 and 10%) are shown in Figure 6. For Mag-CTA^+^, it is easy to define the silicate layers by the dark and bright lines. The nanocomposites (Ros-Lim/Mag-CTA^+^ 10 and 5%) show an intercalated structure of the modified clay. However, the nanocomposite (Ros-Lim/Mag-CTA^+^ 1%) shows a partial or total exfoliated structure, and the clay nanoparticles are mainly well dispersed in the Ros-Lim copolymer matrix. These results confirm the results obtained by XRD analysis.

Figure 7 shows the TGA curves of Ros-Lim copolymer and the obtained nanocomposites 1, 5 and 10% wt of Mag-CTA^+^. It can be seen that all the obtained nanocomposites exhibit a one-step weight loss mechanism. The obtained nanocomposites have residual weights similar to organoclay concentration (1, 5 and 10%) and show a high thermal stability up to a degradation temperature about 320 °C with 10% of Mag-CTA^+^, while the degradation temperature of pure Ros-Lim copolymer observed about 280 °C. This gain in stability is due, according to previous work [30], to the formation of a protective carbonized layer. The formation of this layer is favored by the fine dispersion of intercalated or exfoliated particles of clay, which play an inorganic support role [31]. In general, the degradation temperature of the polymers is increased after the incorporation of exfoliated lamellar silicates [32,33,34]. These results show that the thermal stability of the obtained nanocomposites is not only related to the clay amount but is also much more related to the dispersion of the clay in the copolymer matrix (intercalation or exfoliation)—it is related to the surface area between the copolymer matrix and the clay. Such results indicate that the introduction of clay might change the decomposition mechanism of rosin-limonene copolymer under high temperature.

## 4. Conclusions

The effect of the organoclay (Mag-CTA^+^), prepared and used with different ratios, in the synthesis of green nanocomposites Ros-Lim/Mag CTA^+^ is studied. The XRD results indicate that the nanocomposite prepared with 1% wt of Mag-CTA^+^ was exfoliated, and the nanocomposites prepared with 5 and 10% wt of Mag-CTA^+^ were intercalated, leading to an expansion of the interlayer distance between the layers. Thermo-gravimetric results indicate that the nanocomposites present high thermal stability compared with Ros-Lim copolymer (T < 320 °C). This is attributed to the interactions between all the copolymer chains and the organic compounds of the modified clay. The reinforcing effect of the clay in the copolymer is confirmed by increasing the rigidity of the system. The morphology study by SEM and TEM of the obtained nanocomposites confirmed an organization of certain particles, and in other cases a separation in plates made up of montmorillonite layers; this confirms partial or total exfoliation of montmorillonite in the copolymer matrix and formation of the nanocomposites. In general, the results show that CTAB is effective for the preparation of organoclay. In addition, the study showed that it is possible to prepare such green nanocomposites using the Algerian clay (Maghnite) as a nanoparticle.

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
