# Peer review of "Green Nanocomposites from Rosin-Limonene Copolymer and Algerian Clay"

_polymers, 2020, doi:10.3390/polym12091971_

Round 1
Reviewer 1 Report
The article is well written and organized.
- In the revision, the discussion can be improved with more references to international scientific literature.
- The figures can be redrawn to be clearer and look more professional.
- The scale bar should be more visible in the SEM images.
Author Response
We thank the reviewer for their comments and carefuk reading of our manuscript. We have updated the ms in line with these comments.
Reviewer 2 Report
This paper described the nanocomposites consisting of copolymers and Algerian organophilic-clay named Maghnite-CTA+ (Mag-CTA+) with azobisisobutyronitrile. As a result, the author reported that these composites showed a significant improvement in the thermal stability compared to the pure rosin-limonene copolymer. I recommend this paper to be published after minor revision since the reported results were meaningful in this field.
(1) In Introduction part, the related recent references should be more added in the revised text.
(2) In Figure 5, the resolution of SEM image was poor. The author should add the new image in the revised text.
(3) In Figure 3, '-1' in 'cm-1' should be superscripted.
(4) The reason why the thermal stability was enhanced should be explained in detail in the revised text.
Author Response
We thank the reviewer for their comments and careful reading of our comments. We have updated the ms in line with their comments